# A Review of Optical Fiber Sensing Technology Based on Thin Film and Fabry–Perot Cavity

Chaoqun Ma [1,†], Donghong Peng [1,†], Xuanyao Bai [1], Shuangqiang Liu [1,*] and Le Luo [1,2,3,4,*]

1   School of Physics and Astronomy, Sun Yat-Sen University, Zhuhai 519082, China
2   Shenzhen Research Institute of Sun Yat-Sen University, Nanshan, Shenzhen 518087, China
3   State Key Laboratory of Optoelectronic Materials and Technologies, Guangzhou Campus, Sun Yat-Sen University, Guangzhou 510275, China
4   International Quantum Academy, Shenzhen 518048, China
*   Correspondence: liushq33@mail.sysu.edu.cn (S.L.); luole5@mail.sysu.edu.cn (L.L.)
†   These authors contributed equally to this work.

**Abstract:** Fiber sensors possess characteristics such as compact structure, simplicity, electromagnetic interference resistance, and reusability, making them widely applicable in various practical engineering applications. Traditional fiber sensors based on different microstructures solely rely on the thermal expansion effect of silica material itself, limiting their usage primarily to temperature or pressure sensing. By employing thin film technology to form Fabry–Perot (FP) cavities on the end-face or inside the fiber, sensitivity to different physical quantities can be achieved using different materials, and this greatly expands the application range of fiber sensing. This paper provides a systematic introduction to the principle of FP cavity fiber optic sensors based on thin film technology and reviews the applications and development trends of this sensor in various measurement fields. Currently, there is a growing need for precise measurements in both scientific research and industrial production. This has led to an increase in the variety of structures and sensing materials used in fiber sensors. The thin film discussed in this paper, suitable for various types of sensing, not only applies to fiber optic FP cavity sensors but also contributes to the research and advancement of other types of fiber sensors.

**Keywords:** fiber sensor; Fabry–Perot cavity; thin film

## 1. Introduction

Fiber optic sensing technology utilizes the propagation of light signals in optical fibers to detect external physical quantities. When external physical quantities (temperature, humidity, magnetic field, electric field, etc.) change, the characteristic parameters of the optical signal transmitted through fiber (phase, intensity, wavelength, etc.) also undergo corresponding variations. By establishing the relationship between these characteristic parameters and the physical quantity to be measured, the value of the physical quantity can be derived. Compared to other types of sensors, fiber optic sensors possess advantages such as being compact, lightweight, and resistant to electromagnetic interference. These sensors enable measurements to be conducted in extreme environments, including confined spaces, extreme temperatures, and areas with strong electromagnetic interference. Furthermore, fiber optic sensors demonstrate significant advantages in terms of detection sensitivity, resolution, and signal transmission distance. Based on the signal modulation methods used in fiber optic sensors, they can be classified into several categories, including intensity modulation, wavelength modulation, frequency modulation, polarization modulation, and phase modulation. Interferometric fiber optic sensors based on phase modulation have gained widespread adoption due to their exceptional resolution and sensitivity. Nevertheless, the utilization of interferometric fiber optic sensors based on phase modulation demands high standards for the quality of both the light source and the

receiving detection system. Phase modulation-based fiber optic sensors primarily utilize different types of fiber optic interferometers to achieve optical interference. Typical fiber optic interferometers include the Mach–Zehnder (M-Z) interferometer [1,2], Michelson interferometer [3,4], FP interferometer [5,6], and Sagnac interferometer [7,8]. The FP interferometer, which originated in the late 19th century, typically consists of two highly reflective mirrors. The incident light beam undergoes multiple reflections between these mirrors, resulting in multiple-beam interference. Despite the early invention of the FP interferometer, its integration with optical fibers commenced in the 1980s. The fiber optic FP structure, as a crucial component in the development of fiber optic sensing technology, has been subject to in-depth investigations by researchers. Compared to other sensors, the fiber optic FP sensing structure exhibits significant advantages in terms of dynamic range, sensitivity, response speed, and implementation modes. Figure 1 illustrates the annual number of research papers related to FP fiber optic sensors from 1999 to 2023. As observed, there has been a consistent and steady increase in the number of publications in this field in recent years. This upward trend clearly indicates that FP fiber optic sensors have gained significant attention from the academic community, and the research in this area continues to be extensively explored and studied.

Number of Articles

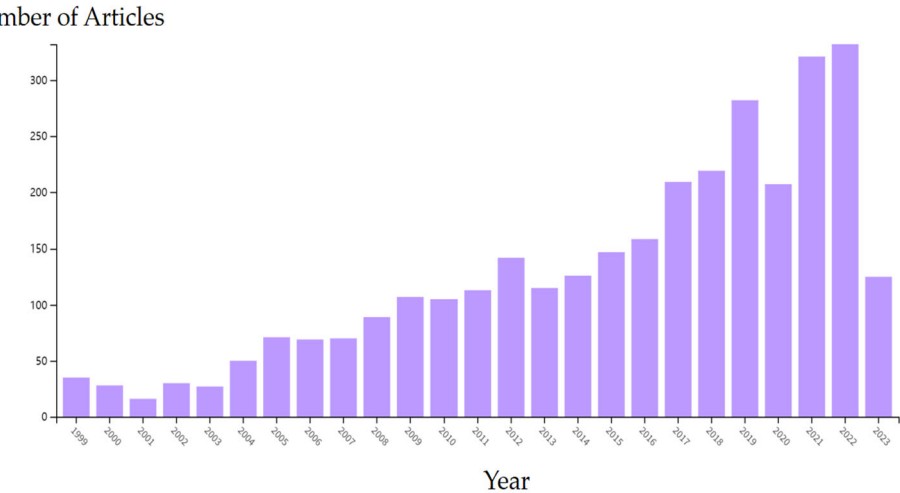

**Figure 1.** Annual numbers of research papers related to FP fiber optic sensors from 1999 to 2023.

In Mach–Zehnder and Michelson interference optical fiber sensors, the elastic-optic effect of the optical fiber is negligible. Therefore, a lengthy optical fiber is necessary for high sensitivity, which in turn, creates poor thermal stability and increased vulnerability to vibration. Additionally, the phase noise produced by the light source has a significant impact on the interferometer structure, requiring a highly coherent light source for optimal sensor performance [9]. Additionally, in the sensing experiment, fiber optic FP interferometric sensors can effectively mitigate optical power fading caused by polarization. Due to the birefringent nature of optical fibers, Mach–Zehnder and Michelson interferometers split a light beam into two independently propagating beams, each with varying polarization states in a random manner. This leads to a decrease in the efficiency of interference when the two beams recombine, resulting in reduced contrast of interference fringes [9]. Sagnac interferometers can employ polarization control techniques or high birefringent fibers to maintain the polarization characteristics of transmitted light, but this undoubtedly adds complexity and cost to the system [10,11]. However, FP interferometers, due to their short cavity length or propagation in air, can effectively ignore the issue of optical power fading [9]. Fiber optic sensors based on FP interferometers have unique structures and can be fabricated through a variety of methods to meet specific sensing requirements. The sensitivity of the sensor can be enhanced by filling the FP cavity with specific materials. Moreover, an open FP cavity can be effectively employed to sense gas pressure or measure the refractive index of solutions. These properties have greatly contributed to the progress

of fiber optic sensing technology. Thin film technology plays a crucial role in fiber optic sensors by coating reflective surfaces inside the FP cavity or on the fiber surface, enabling the measurement of specific physical quantities [12]. For instance, temperature sensing can be achieved by adding a temperature-sensitive film on the reflecting surface [13]. The choice of materials and fabrication methods for the film constituting the FP cavity can be tailored according to the specific requirements of the physical quantity sensor.

Reference [14] summarized the main methods of constructing Fabry–Perot (FP) cavities within optical fibers. These methods can be broadly categorized as non-splicing and splicing techniques. The authors classified important literature on fiber optic FP sensors from 1981 to 2014 based on fabrication methods and sensing applications. They also discussed the development trends in this research direction and provided an overview of successfully implemented fiber optic FP sensors in industrial settings. Reference [15] reported on the research progress of fiber optic sensors based on FP interference. The authors conducted theoretical and experimental analyses of FP sensors in sensing applications such as temperature, displacement, and the refractive indices of liquids and solids. They also provided prospects for future applications of FP sensors in other fields. In contrast to previous review articles, this paper begins with a comprehensive introduction elucidating the principles and sensing mechanisms of fiber sensors employing the FP cavity with thin film. A meticulous analysis is conducted to explore the factors influencing the sensitivity of these sensors. Subsequently, a comprehensive survey is presented, highlighting the research progress of these sensors across diverse domains, encompassing pressure, magnetic field, refractive index, humidity, gas, temperature, as well as biological or medical sensing. Furthermore, an overview is provided, elucidating the diverse thin film materials and FP cavity structures utilized to accomplish these sensing applications.

## 2. Principle

The Fabry–Perot cavity mainly plays a role in changing the spectrum in the fiber optic sensor technology, changing the external parameters. In conventional fiber optic FP interferometers, one of the cavity mirrors is typically formed by a cleaved single-mode fiber (SMF), while the other mirror consists of a mirror parallel to the fiber end-face. This parallel mirror can be a thin film or another section of the fiber end-face. Upon the emission of light from the fiber core, it traverses the fiber end-face and encounters the mirror, leading to multiple reflections and interference. The optical path difference between consecutive reflections is determined as 2nd, and the corresponding phase difference can be expressed as follows:

$$\varphi = \frac{4\pi n d}{\lambda} \tag{1}$$

where $n$ represents the refractive index of the medium between the two cavity mirrors, $d$ denotes the separation distance between the two mirrors, and $\lambda$ signifies the wavelength of the incident light wave.

Considering an incident light wave with an amplitude of $E$ and an initial phase factor of 0, the fiber end-face is characterized by transmission coefficient $t_1$ and reflection coefficient $r_1$. In the counter-propagating direction, the transmission coefficient is denoted as $t_1'$, and the reflection coefficient is represented as $r_1'$. Additionally, the mirror surface exhibits a reflection coefficient denoted as $r_2$. Within the fiber core, all reflected light waves experience interference, resulting in a complex amplitude expressed as [16]:

$$
\begin{aligned}
E_r &= r_1 E e^0 + t_1 t_1' r_2 E e^{-i\varphi} + t_1 t_1' r_1' r_2^2 E e^{-i2\varphi} + \cdots + t_1 t_1' r_1'^{m-1} r_2^m E e^{-im\varphi} \\
&= \left[ r_1 E + t_1 t_1' r_2 E e^{-i\varphi} \left( 1 + r_1' r_2 e^{-i\varphi} + \cdots + r_1'^{m-1} r_2^{m-1} e^{-i(m-1)\varphi} \right) \right] \\
&= r_1 E + t_1 t_1' r_2 E e^{-i\varphi} \frac{1 - r_1'^m r_2^m e^{-im\varphi}}{1 - r_1' r_2 \eta e^{-i\varphi}}
\end{aligned}
\tag{2}
$$

where $m$ represents the order of the reflected light. The parentheses in the equation denote the sum of a geometric series. When $m$ tends to infinity, we obtain:

$$E_r = r_1 E + t_1 t_1' r_2 E e^{-i\varphi} \frac{1}{1 - r_1' r_2 e^{-i\varphi}} = \frac{r_1 E + (t_1 t_1' - r_1' r_1) r_2 E e^{-i\varphi}}{1 - r_1' r_2 e^{-i\varphi}} \tag{3}$$

according to Fresnel equations:

$$\begin{aligned} r_1^2 = r_1'^2 = R \\ t_1 t_1' = 1 - r_1^2 = T \end{aligned} \tag{4}$$

the total optical intensity of the fiber-reflected FPI is:

$$I_R = E_r \cdot E_r^* = I_0 \cdot \frac{R_1 + R_2 - 2\sqrt{R_1 R_2}\cos\varphi}{1 + R_1 R_2 - 2\sqrt{R_1 R_2}\cos\varphi} \tag{5}$$

In the scenario where the reflectivity $R$ is extremely low, Equation (5) can be simplified to the following form:

$$I_R = 2R(1 - \cos\varphi)I_0 \tag{6}$$

the transmitted light intensity is given by the following expression:

$$I_T = I_0 - I_R = I_0[1 - 2R(1 - \cos\varphi)] \tag{7}$$

It can be observed that the magnitude of the above expression primarily depends on the phase difference $\varphi$. When:

$$\varphi = \frac{4\pi n d}{\lambda_k} = 2k\pi \tag{8}$$

The interference of the reflected light results in destructive interference, characterized by the positive integer $k$ representing the order of resonant wavelengths. Hence, we can express it as follows:

$$\lambda_k = \frac{2nd}{k} \tag{9}$$

differentiating both sides of Equation (9), we obtain:

$$\Delta\lambda_k = \lambda_k \left( \frac{\Delta n}{n} + \frac{\Delta d}{d} \right) \tag{10}$$

Herein, $\Delta\lambda_k$ represents the variation in resonant wavelength, $\lambda_k$ signifies the resonant wavelength, $\Delta n$ denotes the change in the refractive index of the medium, and $\Delta d$ represents the alteration in cavity length. Consequently, modifications in the cavity length $d$ or refractive index $n$ within the cavity induce shifts in the wavelength where destructive interference transpires, thus causing alterations in the interference spectrum. Perturbations in external environmental factors, such as temperature, humidity, or magnetic field, prompt deformations in the thin film, ultimately influencing the FP cavity length or the refractive index of the medium within the cavity. Consequently, the resonant wavelength experiences changes. By measuring the shift in resonant wavelength, the corresponding variation in the external environmental parameter can be determined. Through experimental investigations that establish the relationship between the shift in resonant wavelength and the corresponding environmental parameter, the detection of wavelength shifts enables the inference of changes in the environmental parameter. Thus, the sensing objective can be accomplished.

The aforementioned formula applies to cases where the thin film is extremely thin, rendering its thickness negligible. In this scenario, only two reflecting surfaces need to be considered. When a thin film of a certain thickness is used as the reflecting film in the FP cavity, the consideration of three reflecting surfaces is necessary, as shown in Figure 2. The

interfaces $M_1$ and $M_2$ form $FP_1$, while $M_2$ and $M_3$ form $FP_2$. The interface reflectance ($R$) is given by the following equation:

$$R_1 = \left(\frac{n_1 - n_2}{n_1 + n_2}\right)^2$$
$$R_2 = R_3 = \left(\frac{n_3 - n_2}{n_3 + n_2}\right)^2 \tag{11}$$

where $n_1$, $n_2$, and $n_3$ represent the refractive indices of the fiber, air, and the thin film, respectively.

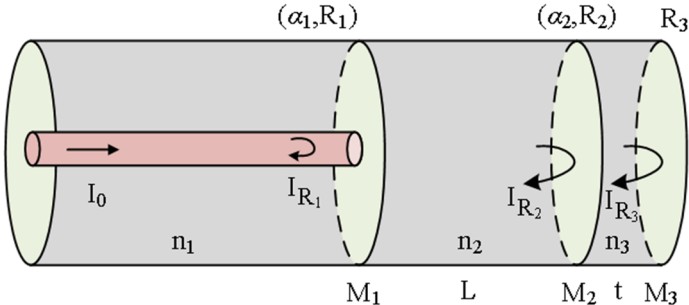

**Figure 2.** Description of the Optical Field in Thin Film-Based Fiber FP Cavity.

In this analysis, we account for the interference of three reflected beams within the SMF while disregarding higher-order terms. By considering the electric field amplitudes and phase factors, the corresponding expression can be derived as follows:

$$E_r = \sqrt{R_1}E_0 + (1 - \alpha_1)(1 - R_1)\sqrt{R_2}E_0 e^{-j2\varphi_1}$$
$$+ (1 - \alpha_2)(1 - \alpha_1)(1 - R_2)(1 - R_1)\sqrt{R_3}E_0 e^{-j2(\varphi_1 + \varphi_2)} \tag{12}$$

where $E_r$ represents the total amplitude of the interference field, $E_0$ represents the initial amplitude of the incident field, and $\alpha$ represents the optical transmission loss coefficient. The phase factors $\varphi_1$ and $\varphi_2$ are given by:

$$\varphi_1 = \frac{4\pi n_2 L}{\lambda}, \varphi_2 = \frac{4\pi n_3 t}{\lambda} \tag{13}$$

By representing the power of the FP reflected light as the ratio between the reflected electric field intensity and the incident electric field intensity, the following expression can be derived [17]:

$$
\begin{aligned}
I_r(\lambda) =& |E_r / E_0|^2 \\
=& R_1 + (1 - \alpha_1)^2(1 - R_1)^2 R_2 \\
&+ (1 - \alpha_1)^2(1 - \alpha_2)^2 \times (1 - R_1)^2(1 - R_2)^2 R_3 \\
&+ 2\sqrt{R_1 R_2}(1 - \alpha_1)(1 - R_1)cos(\varphi_1) \\
&+ 2\sqrt{R_2 R_3}\left[ \begin{array}{c} (1 - \alpha_1)^2(1 - \alpha_2)(1 - R_1)^2(1 - R_2) \times cos(\varphi_2) \\ + 2\sqrt{R_1 R_3}(1 - \alpha_1)(1 - \alpha_2)(1 - R_1) \times (1 - R_2)cos(\varphi_1 + \varphi_2) \end{array} \right]
\end{aligned} \tag{14}
$$

It can be observed that the reflected power in the above equation is primarily composed of three cosine functions that are linearly superimposed. Considering that M1 and M2 form FP1, M2 and M3 form FP2, and M1 and M3 form FP3, each cosine term corresponds to the optical path difference of FP1, FP2, and FP3 cavities, respectively. By employing bandpass filtering techniques, the signals corresponding to different FP cavities can be extracted from the overall spectrum. Typically, when external parameters such as temperature or pressure change, the deformation of the thin film causes a variation in the length of FP1, leading to a spectral shift in the interference pattern associated with FP1. By measuring the displacement of the resonant wavelength, the corresponding change in the environmental parameter can be determined.

When evaluating the performance of a sensor, it is important to consider not only sensitivity but also other parameters, such as the Figure of Merit and Limit of detection. The Figure of Merit (FOM) is an important metric for directly quantifying sensor performance. It is defined as the ratio of the sensor's sensitivity and FWHM, where FWHM is full width at half-maximum of the interference spectrum [18]. As for Fabry–Perot interference, its full width at half maximum (FWHM) is defined as follows [19]:

$$FWHM = \frac{2(1-R)}{\sqrt{R}} \tag{15}$$

Here, $R$ is the film reflectance. It is evident that to enhance the FOM of a sensor, we can either improve its sensitivity or increase thin film reflectance. The Limit of Detection (LOD) for a sensor is the minimum detectable change in the measured physical quantity. When using spectral shift, the LOD of a sensor is calculated by dividing the wavelength resolution by the sensor's sensitivity [20]. In order to enhance LOD, it is recommended to utilize a spectrometer with greater spectral resolution, opt for a consistent light source, and mitigate environmental disturbances.

It should be noted that in addition to the conventional Fabry–Pérot (FP) cavity mentioned above, there are other FP-like structures in optical fibers, such as the anti-resonant reflecting optical waveguide (ARROW). Proposed by Duguya et al. in 1986 [21], ARROW emerged to address the issue of electromagnetic wave energy leakage caused by the significant difference in refractive index between traditional optical waveguides and silicon-based waveguides. Similar to the FP cavity, ARROW employs a higher refractive index layer between the waveguide and the substrate to confine the propagation of light beams in the waveguide layer. The ARROW structure not only reduces the energy leakage but also elevates the percentage of the energy distribution of the evanescent field in the material being evaluated, enhancing the sensitivity of the waveguide to changes in external physical quantities. Initially, ARROW's application was focused on the field of silicon-based waveguides, with little cross-section with the optical fiber. However, as the photonic crystal fiber preparation technology has matured and improved, ARROW has gradually entered the research scope of fiber sensing. Reference [22] has conducted theoretical analysis and discussion on the anti-resonant mechanism in photonic crystal waveguides, proving the anti-resonant interference phenomenon of light in hollow-core photonic crystal fibers. This provides significant insight for the future application of ARROW in photonic crystal fibers and also opens up a new path for the integration of single-layer quartz tubes and fiber sensing areas. Hence, for integrated waveguide optical sensing, ARROW offers excellent sensing performance and application prospects [23,24].

## 3. Research Progress of Optical Fiber Fabry–Perot Cavity Sensors

Fiber optic FP cavity sensing is a critical research field that has witnessed significant advancements in recent years. This technology utilizes the FP cavity structure within an optical fiber to achieve high-sensitivity detection of environmental parameters by monitoring the phase variation of the light signal. These sensors offer advantages such as simple structure, high sensitivity, and rapid response, making them widely applicable in fields such as pressure, magnetic field, refractive index, humidity, gas detection, temperature, and biomedical applications. By optimizing the FP cavity structure and adjusting the types of thin film employed, the performances of these sensors can be further enhanced. The research on fiber optic FP cavity sensors and thin film provides essential technical support for achieving high-precision and high-sensitivity monitoring of environmental parameters, opening up new possibilities for scientific research and engineering applications across various domains.

### 3.1. Pressure Sensor

Pressure measurement is crucial in diverse domains, including oceanography, geology, and medical applications. Notably, significant progress has been made in recent years

with the development of various fiber optic pressure sensors. These sensors employ different principles, such as Fiber Bragg Gratings (FBGs) [25,26], Surface Plasmon Resonance (SPR) [27,28], Multimode Interference (MMI) [29,30], and Mach–Zehnder interferometer (MZI) [31,32]. This section primarily focuses on recent advancements in fiber optic FP pressure sensors utilizing thin film technology.

Silicon dioxide ($SiO_2$) thin diaphragms possess ideal attributes such as high temperature resistance, structural stability, and chemical inertness [33], rendering them suitable for measurements in extreme environments. In [34], a compact fiber optic FP pressure sensor based on a silicon diaphragm was proposed, as shown in Figure 3. The authors utilized a carbon dioxide laser to weld an SMF to a capillary tube, creating a sealed FP cavity between the end-face of the SMF and the silicon diaphragm on the capillary tube's end-face. When the external pressure changes, the silicon diaphragm undergoes deformation, altering the cavity length of the FP cavity. This change in the optical path difference between the two end-faces results in a shift in the interference spectrum, achieving a sensitivity of 9.48 pm/kPa. Additionally, an alternative approach was proposed by [35,36]. This method entailed etching a groove at the fiber end using hydrofluoric acid, followed by bonding the silicon diaphragm to the fiber end through the application of heat. Consequently, FP cavities were formed, exhibiting sensitivities of 11 nm/kPa and 12.4 nm/kPa, respectively.

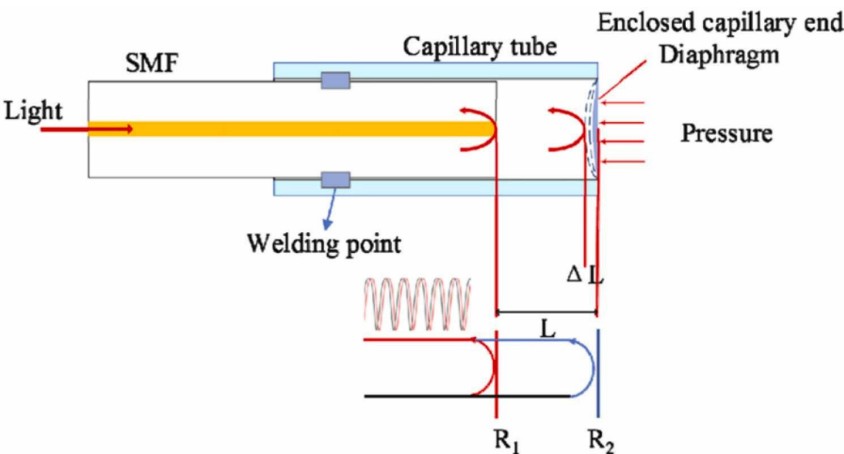

**Figure 3.** Formation of an FP cavity using an SMF and a capillary tube [34].

However, both silicon and silicon dioxide possess a high Young's modulus [37]. Typically, optical fiber dimensions are around one hundred micrometers, and even with the optimal thickness of the silicon diaphragm, achieving high sensitivity in such dimensions remains challenging. In recent years, considerable research interest has been directed toward the utilization of polymer film for pressure sensing. This film offer advantages such as a low Young's modulus, cost-effectiveness, and flexibility [38,39]. Among the various options, polydimethylsiloxane (PDMS) stands out as a polymeric organosilicon compound. Solid PDMS represents a transparent elastomer that can be easily and swiftly processed. It is cost-effective, optically transparent, and readily bonded with different materials at room temperature [40]. The remarkable elasticity of PDMS can be attributed to its low Young's modulus [41], rendering it an excellent choice as a thin film material for pressure-sensing applications. Reference [42] proposed a fiber optic pressure sensor utilizing PDMS as the reflective film for the FP cavity, as depicted in Figure 4. The authors fused a glass tube to the end of an SMF, followed by the deposition of a PDMS film layer at the end of the glass tube. The incident light beam undergoes the first reflection at $R_1$ (end-face of the SMF) and the second reflection at $R_2$ (left side of the PDMS film) after passing through the FP gas chamber. Subsequently, it undergoes the third reflection at R3 (the right side of the PDMS film). These three surface-reflected beams interfere inside the SMF. When external pressure increases, the polymer film undergoes deformation, resulting in a change in the FP cavity length and a shift in the interference spectrum. Compared to similar research

works on PDMS-based pressure sensors, the authors significantly reduced the thickness of the film. This reduction not only enhances the film's response to external pressure but also effectively mitigates the impact of temperature on the pressure sensor. Experimental data show that the pressure sensitivity of the sensor reaches 100 pm/kPa. Similar studies on PDMS-based pressure sensors include [43], which filled a hollow-core fiber with PDMS film at the fiber end to form an FP cavity for measuring gas pressure, achieving a pressure sensitivity of 52.143 nm/MPa. In references [44,45], PDMS film was filled in a capillary tube to form an FP cavity with the end of an SMF for gas pressure measurement, yielding pressure sensitivities of 20.63 nm/MPa and 55 pm/mBar, respectively.

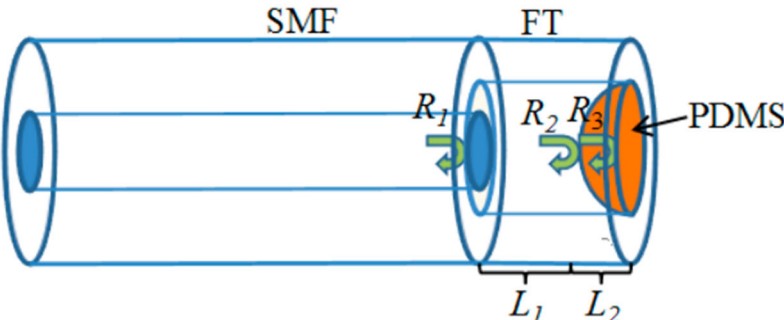

**Figure 4.** Fiber FP pressure sensors based on PDMS thin film [42].

The fabrication of PDMS film possesses challenges in controlling its thickness and achieving uniform symmetry. Ultraviolet curable polymers are a promising candidate for pressure-sensing film due to their ability to solidify rapidly under ultraviolet (UV) light exposure without requiring high-temperature curing [46,47]. In reference [48], UV-curable polymer film was coated on the end of a hollow core silica tube, forming an FP cavity. The film deforms under external pressure, resulting in a change in cavity length and a corresponding shift in the interference spectrum. Experimental results demonstrated a pressure sensitivity of approximately 396 pm/kPa within the range of 0 to 30 kPa. The thickness of the polymer film is also a crucial factor influencing the sensing performance. To address this, the authors employed a suspension curing method to facilitate precise control over the formation of the UV-curable polymer layer, enabling easy regulation of the film thickness. Additionally, reference [49] proposed a fiber optic FP sensor for perceiving sound pressure using a UV adhesive diaphragm. In this sensor configuration, an FP cavity is formed between the end of an SMF and the UV adhesive diaphragm. The sensing principle is similar to the previous approach. The authors obtained the sound pressure signal by demodulating the changes in reflected optical intensity. Experimental results indicate a sensitivity of 57.3 mV/Pa and a detection range of 21.4 mPa to 3.56 Pa for the sound pressure sensor at a frequency of 1000 Hz. In reference [50], the author employed a welding technique to attach a quartz glass tube to the end of an SMF. Subsequently, a film ultra-thin microbubble structure was created on a liquid-state AB epoxy glue using a bubble-blowing method. Once the bubble solidified, it was transferred to the end of the quartz glass tube, thereby establishing an FP cavity in conjunction with the SMF. Based on the experimental data, the pressure sensor exhibited a sensitivity of 263.15 pm/kPa within the pressure range of 100.0 kPa to 400.0 kPa.

In addition to the devices that utilize polymer as reflective film, there is another approach where polymers are applied externally to the fiber to enhance its pressure-sensing properties. Reference [51] presented an optical fiber FP sensor for measuring water pressure variations. The pressure sensor consisted of two SMFs and an interposed portion of a photopolymerizable resin between their end-faces, forming an FP cavity. When the external water pressure changes, the resin undergoes strain (the study's analysis suggests that the physical length change of the polymer contributes more significantly than the refractive index change), leading to a variation in the FP cavity length and a corresponding shift in the interference spectrum. The study indicates that with an increase in the number of

polymer resin layers coated on the outside of the FP cavity, the response of the FP cavity length to strain becomes larger. Consequently, the static hydrostatic pressure sensitivity of the fiber FP pressure sensor increases from 102 to 475 pm/MPa, and its detection range spans from 0 to 20 MPa.

Using thinner film layers can effectively enhance the sensitivity of sensors. Graphene, the thinnest material discovered to date, with a thickness as low as around 0.3 nm [52], exhibits exceptionally high mechanical strength [53]. Utilizing graphene as a thin film can greatly improve the pressure sensitivity of sensors. Reference [54] presented a setup where a capillary tube was fusion-spliced to the end of an SMF, and a graphene film with a thickness of approximately 0.71 nm was transferred to the other end of the capillary tube, forming an FP cavity with the end-face of the SMF. Experimental results demonstrated an average pressure sensitivity of 39.4 nm/kPa within the range of 0–5 kPa. However, the authors observed slight gas leakage into the cavity during the experiments, leading to some measurement errors. Reference [55] proposed a similar pressure sensor based on graphene film. In this setup, the end-face of an SMF was etched to create a groove, and a graphene film was transferred onto the fiber end-face. Experimental results showed a response of 1.28 nm/mmHg for pressure variations in the range of 0–100 mmHg. The authors also provided theoretical analysis for such sensors and proposed a critical thickness for the film. When the film thickness is below this threshold, the sensitivity of the sensor is no longer influenced by the film's thickness and elastic properties, and the performance can be enhanced through FP cavity design. Similarly, in reference [56], an FP cavity formed by the end-face of an SMF and graphene film is utilized for sensing changes in sound pressure, achieving a sensitivity of 43.5 dB re 1 V/Pa at 60 Hz. In reference [57], an FP cavity is formed by the end-face of a multimode fiber and a graphene film for pressure sensing, achieving a sensitivity of 79.956 nm/kPa.

Compared to the aforementioned materials, metal film offers unique advantages due to its high reflectivity and ideal properties, such as being lightweight, physically stable, and easy to manufacture. In reference [58], the authors employed a ceramic ring as a cavity and constructed an FP interferometer using SMF end-faces and a silver film at both ends. When the silver-coated sensing membrane is deformed by external pressure, it causes a change in the length of the FP cavity. The experimental result presented by the authors in the paper is the response of cavity length to external pressure. The sensitivity of cavity length variation for this sensor within the range of 0–10 mPa is reported to be 1677 nm/MPa, with a stress measurement resolution of 60 Pa. In a similar manner, reference [59] also proposed an FP structure enhanced with a silver film with a static pressure sensitivity of 1.6 nm/kPa.

Gold film can also be employed as pressure-sensitive membranes, offering greater chemical stability compared to silver film. Reference [60] proposed a fiber optic FP pressure sensor with a gold film. In this study, the authors placed an SMF and a gold film at the ends of a ceramic ferrule to form the FP cavity, following the same sensing principle as the silver film. Experimental results demonstrated that within the range of 0–100 kPa, this method exhibits a static pressure sensitivity of approximately 19.5 nm/kPa. FP fiber optic pressure sensors have been commercially applied in various examples. For instance, Roctest Company offers the FOP series (FOP-F, FOP-C, and FOP-P) fiber optic piezometers based on stainless steel diaphragms, primarily used in civil engineering. These sensors provide accurate and reliable measurements unaffected by electromagnetic, RF, and lightning interference. Additionally, FISO's FOP-M sensor is designed for pressure measurement under high-temperature conditions, such as in aerospace and automotive research and development, where harsh and hazardous environments are common. The FOP-M pressure sensor exhibits resistance to EMI/RFI/MW, compact size, and reliable measurements in adverse conditions. It boasts good operating conditions, high accuracy, and resistance to corrosive environments. The performances of fiber optic FP pressure sensors based on different thin films, as discussed in this section, are summarized in Table 1.

**Table 1.** Pressure sensor performance based on different materials.

| Material | Sensitivity | Test Range | Reference |
|---|---|---|---|
| Silicon | 9.48 pm/kPa | 0~200 kPa | [34] |
| | 11 nm/kPa | 0~100 kPa | [35] |
| | 12.4 nm/kPa | 6.9~48.3 kPa | [36] |
| PDMS | 100 pm/kPa | 100~175 kPa | [42] |
| | 52.143 nm/Mpa | 0.1~0.7 Mpa | [43] |
| | 20.63 nm/Mpa | 0~2 MPa | [44] |
| | 55 pm/mBar | 0~50 mBar | [45] |
| UV | 395 pm/kPa | 0~30 kPa | [48] |
| | 57.3 mV/Pa | 21.4 mPa~3.56 Pa | [49] |
| AB epoxy glue | 263.15 pm/kPa | 100.0 ~400.0 kPa | [50] |
| Graphene | 39.4 nm/kPa | 0~5 kPa | [54] |
| | 1.28 nm/mmHg | 0~100 mmHg | [55] |
| Silver | 1.6 nm/kPa | 0~50 psi | [59] |
| Gold | 19.5 nm/kPa | 0~100 kPa | [60] |

*3.2. Magnetic Field Sensor*

Magnetic field measurement is very important in various fields, such as scientific research and industrial production. Due to the excellent electromagnetic interference resistance of optical fibers, they have proven to be outstanding magnetic field sensors [61]. Currently, the commonly used material for fiber optic magnetic field sensing is magnetic fluid, which is a composite material formed by suspending magnetic particles in a liquid. These magnetic particles are typically made of ferrite or metallic materials such as iron, nickel, and cobalt, while the liquid base can be water or organic solvents. Under the influence of an external magnetic field, the magnetic fluid particles are guided by magnetic forces, forming chain-like, cluster-like, or ordered structures, thereby altering the morphology and properties of the magnetic fluid. When the external magnetic field disappears, the magnetic fluid particles return to a freely suspended state [62,63]. In recent years, there has been significant development of fiber optic sensors utilizing the properties of magnetic fluids, such as magnetic field sensors based on the MZI [64,65], the Michelson interferometer [66,67], and the multimode interference [68,69]. A wide range of fiber optic magnetic field sensors based on FP Interferometers have been proposed. In reference [70], a reflective FP magnetic field sensor based on the Magneto-Volume Effect of Magnetic Fluid was presented. The authors constructed a fiber optic FP cavity by enclosing two segments of single-mode fibers (SMFs) within a capillary. Subsequently, a precisely measured quantity of magnetic fluid was introduced into the cavity, while a specific section of the cavity was intentionally left empty. As the incident light sequentially traversed the SMF and the empty cavity, it eventually reached the interface of the magnetic fluid (MF), encountering two reflective surfaces: the SMF interface and the MF interface. The reflected light from these two interfaces interfered with the SMF. When an axial magnetic field was applied to the sensor, the magnetic fluid underwent deformation, resulting in changes in the lengths of the empty cavity which served as the FP cavity. Experimental results demonstrated that within the range of 15.5–139.7 G, the sensitivity of this magnetic field sensor reached 268.81 pm/G. Under a step-change magnetic field, the sensor exhibited a rapid response time of 0.2 s. Another similar FP cavity based on the Magneto-Volume Effect of Magnetic Fluid was reported in reference [71]. The authors fusion-spliced an SMF with an HCF and filled the HCF with a controlled amount of magnetic fluid, leaving a section of air column. The interference principle is the same as described above. Experimental results showed that within the range of 109.6–125.8 G, the sensitivity of this sensor was −4219.15 pm/G. However, within the broader range of 0–125.8 G, the linearity of the sensor's response was not ideal, leading to a limited sensing range.

Apart from experiencing changes in volume, magnetic fluid also demonstrates alterations in refractive index due to fluctuations in the magnetic field. Reference [72] proposed a fiber optic FP magnetic field sensor based on the refractive index effect of magnetic fluid. The sensor's FP cavity consisted of an SMF, a capillary glass tube, and the SMF end-face coated with a gold film. The authors inserted the SMF into the capillary tube, leaving a section of air column in the middle, and then encapsulated it with another SMF coated with a gold film. Magnetic fluid was filled in between the capillary tube and the SMF. Due to capillary action [73], the surface of MF became curved and appeared concave. Moreover, the length of the cavity containing the magnetic fluid was very small, making the impact of volume changes negligible. Under the influence of an external magnetic field, the refractive index of the magnetic fluid changed, causing the interference spectrum to shift. Experimental results demonstrated that within the range of 118.768~166.261 G, the sensitivity of the sensor can reach 1.02602 nm/G.

Due to the high viscosity of the magnetic fluid, its response time to magnetic fields is slow, and it is prone to leakage. Researchers have begun investigating the utilization of polymer materials filled with magnetic particles as an alternative to magnetic fluid for sensing applications. Magnetic-Based Polydimethylsiloxane is a magnetic-sensitive polymer formed by filling ferromagnetic material ($Mn_3O_4$ nanocrystals) into PDMS. In reference [74], the authors used this material as a thin film inside the FP cavity for dual parameter sensing of temperature and magnetic fields. The magnetic-sensitive polymer is coated at the end of an SMF, which forms an FP cavity with another SMF end-face. When there are changes in the external magnetic field or temperature, the magnetic-sensitive material undergoes deformation, causing a change in the FP cavity length and leading to a shift in the interference spectrum. The linear sensitivity of this sensor within the linear magnetic field intensity range of 0–4 mT is 563.2 pm/mT. Reference [75] proposed a fiber optic FP magnetic field sensor based on a magnetic alloy, as shown in Figure 5a. The FP cavity was composed of Fiber Bragg Gratings (FBGs) in two segments of the SMFs, which were filled with silicone between the two SMFs. The device was then bonded to a magnetic alloy called iron–cobalt–vanadium Supermendur using UV glue. This magnetic alloy has a high saturation magnetic flux density (2.4 T) and extremely low magnetic field sensitivity, making it suitable for sensing in a wide range but not for situations with small magnetic fields. When there was a change in the external magnetic field, the magnetic alloy underwent deformation. Due to the lower Young's modulus of silicone compared to silica, the length of the FP cavity changed while the FBG experienced no strain. With the help of this FP structure, its performance could be significantly improved, enabling measurements in the presence of small magnetic fields. The experimental results indicated that the sensitivity of this sensor within the range of 0–70 mT is −34.83 pm/mT. Reference [76] proposed a magnetic field sensor based on magnetostrictive materials. The FP cavity was formed by fusion splicing a segment of hollow-core fiber between two segments of SMFs, and it was fixed onto a magnetostrictive material called Terfenol-D. This material underwent strain when the magnetic field changed, resulting in a change in the length of the FP cavity. However, the response of this material to the magnetic field was nonlinear within a wide range, and Terfenol-D ceased to respond when the magnetic field intensity exceeded 60 mT. Experimental data showed that the sensitivity within the linear range of 10–30 mT is 14.6 pm/mT. Similar approaches using Terfenol-D for magnetic field sensing are also described in [77], where the authors' fusion-spliced a segment of hollow-core fiber between an SMF and a hollow silica capillary, and Terfenol-D was attached, as shown in Figure 5b. Experimental data showed a sensitivity of −7.53 nm/mT within the range of 4–10 mT. The performance of the magnetic field sensors described in this section is summarized in Table 2.

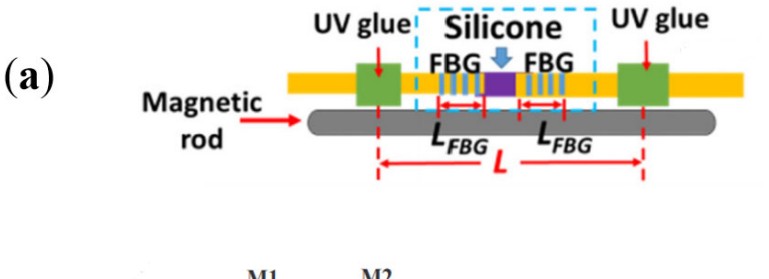

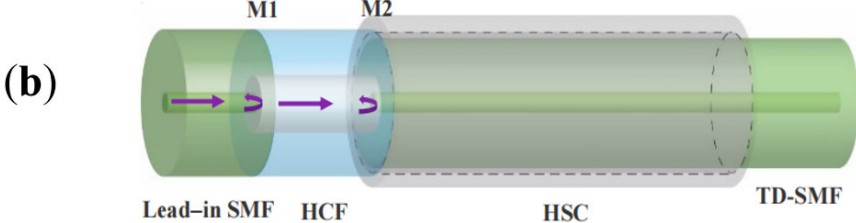

**Figure 5.** (**a**) Magnetic field sensors based on magnetic alloys [75] (**b**) Magnetic field sensors based on Terferol-D [77].

**Table 2.** Magnetic field sensor performance based on different materials.

| Material | Sensitivity (pm/mT) | Test Range (mT) | Reference |
|---|---|---|---|
| MF | 2688.1 | 1.55~13.97 | [70] |
| | −42,191.5 | 10.96~12.58 | [71] |
| | 10,260.2 | 11.8768~16.6261 | [72] |
| $Mn_3O_4$-PDMS | 563.2 | 0~4 | [74] |
| Magnetic alloy | −34.83 | 0~70 | [75] |
| Terfenol-D | 14.6 | 10~30 | [76] |
| | −7530 | 4~10 | [77] |

*3.3. Refractive Index Sensing*

In the fields of biology, chemistry, and materials science, refractive index measurement plays a crucial role. Various optical fiber sensors are employed for refractive index measurement, including those based on the whispering gallery mode (WGM) interference [78,79], MZI [80,81], surface plasmon resonance (SPR) [82,83], and Bragg fiber gratings [84,85]. To measure the refractive index of a solution, a semi-open Fabry–Perot (FP) cavity can be employed, with a thin film coated on the end-face of the cavity to enhance reflectivity and ensure a high contrast of interference fringes. Reference [86] proposed an optical fiber FP refractive index sensor. A cavity was etched on the end-face of an SMF using laser ablation to create the FP cavity. SiO/TiO film was coated on the reflective interface of the FP cavity as mirrors. A microchannel was present at the fiber end, serving as an inlet for the tested liquid or gas. It is necessary for the microchannel to be sufficiently small to maintain the optimal performance of the FP cavity. When the liquid or gas entered through the small hole, it changed the refractive index in the FP cavity, resulting in a shift in the interference spectrum. The experimental results demonstrated a high sensitivity of 1130.887 nm/RI, indicating the sensor's capability for accurate refractive index measurements. Furthermore, by adjusting the cavity length or enhancing the reflectivity, the sensor response and optical performance can be effectively optimized. A similar refractive index sensor was proposed in reference [87]. In this case, a silver film was used as the reflective coating. The measured refractive index sensitivity is 1025 nm/RIU. Reference [88] suggested the use of ZnO coated on both ends of an SMF as a thin film to form the FP cavity for refractive index measurement of chemical substances. Reference [89] introduced a fiber refractive index sensor, where two SMF ends formed an FP cavity, which was subsequently filled with UV glue. This sensor responded to changes in the refractive index of the liquid based on the evanescent field effect [90], with a refractive index sensitivity of 156.8 nm/RIU. Reference [91] presented a

simple fiber FP sensor with a structure similar to the previous one. The authors filled the FP cavity with three different polymers, UV88, NOA68, and Loctite 3525. The refractive index sensing performance of these three materials is shown in the table below, along with the refractive index sensing performance of other refractive index sensors mentioned in this section (Table 3).

**Table 3.** Refractive index sensor performance based on different materials.

| Material | Sensitivity (nm/RIU) | Reference |
|----------|---------------------|-----------|
| SiO/TiO | 1130.887 | [86] |
| Silver | 1025 | [87] |
| UV | 156.8 | [89] |
| UV88 | 24.678 | [91] |
| NOA68 | 81.096 | [91] |
| Loctite 3525 | 34.395 | [91] |

*3.4. Humidity Sensor*

With increasing attention to meteorology, agriculture, industry, and indoor environments, the importance of humidity sensors has become more prominent. Fiber optic humidity sensors have been proposed, such as humidity sensors based on FBG [92,93], long-period fiber gratings (LPFG) [94,95], MZI [96,97], and Michelson interferometers [98,99]. Polymethyl methacrylate (PMMA) is a commonly used synthetic polymer, also known as acrylic or organic glass, which can absorb or release moisture, causing volume changes. When the humidity increases, PMMA absorbs water molecules and expands, resulting in an increase in volume. When the humidity decreases, PMMA releases water molecules and contracts. This property can be utilized to detect humidity changes and is less susceptible to external environmental interference [100]. Reference [101] proposed an optical fiber humidity sensor based on an external FP interferometer structure. The authors fusion-spliced a section of coreless fiber (CLF) to the end of an SMF to enlarge the beam diameter and improve the stability of the extinction ratio of the interference spectrum. The end-faces of CLF and another SMF formed an FP cavity, which was filled with PMMA. When the external humidity changed, PMMA underwent deformation, causing a change in the FP cavity length and resulting in a shift in the interference spectrum. Experimental results showed that within the range of 25% to 80% relative humidity (RH), the RH sensitivity was 0.1747 nm/%RH, with a response time of 4.5 min, and the sensor was unaffected by temperature change in the range of 30–55 °C. Reference [102] utilized a multimode fiber end-face and an SMF end-face to form an FP cavity and filled the cavity with PMMA for humidity sensing. Within the range of 35% to 85% RH, the sensor achieved a sensitivity of 127 pm/%RH. Reference [103] wrapped PMMA material around the end of an SMF to form an FP cavity for humidity sensing. Within the range of 10% to 70% RH, the sensor achieved a sensitivity of 0.4172 nm/%RH.

There are other kinds of polymers that can be used for humidity sensing. Reference [104] proposed an open FP cavity, which was filled with nanocomposite poly-acrylamide (PAM). PAM exhibits a change in refractive index when it absorbs water molecules [105]. A change in external humidity causes a shift in the interference spectrum output by the sensor. Experimental results showed that the relative humidity (RH) sensitivity of this sensor was approximately 0.1 nm/%RH in the range of 38% to 78% RH, and approximately 5.868 nm/%RH in the range of 88% to 98% RH. Reference [106] suggested using two Bragg fiber gratings as an FP cavity and filling the gap between the two Bragg fiber gratings with agarose gel. This sensor exploited the property of agarose gel to expand and change the refractive index with increasing humidity [107]. Moreover, FBGs are sensitive to temperature, which effectively solves the problem of cross-sensitivity between temperature and humidity. Within the range of 43% to 63% RH, the average sensitivity of this sensor was 22.5 pm/%RH. Reference [108] proposed filling the gap between two SMF end-faces with poly(N-isopropyl acrylamide) (PNIPAM) hydrogel, as shown in Figure 6. This sensor

utilized the refractive index change property of PNIPAM with humidity variation [109]. In the relative humidity range of 45% to 75%, the measured relative humidity sensitivity was 1.634 nm/%RH, with good repeatability.

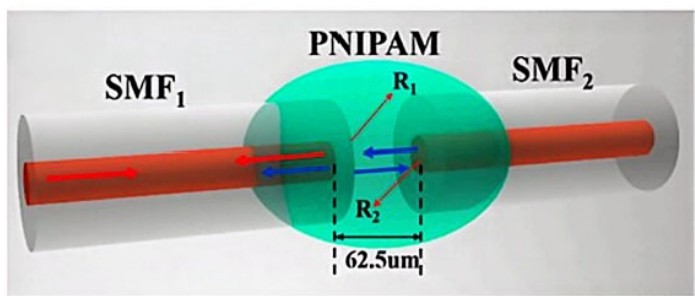

**Figure 6.** Humidity sensors based on the use of PNIPAM [108].

Using polymers as reflective film in the FP cavity for humidity sensing is one of the main techniques used today. In reference [110], a sensor for simultaneous measurement of temperature and relative humidity was proposed, consisting of an FBG and an FP interferometer. The author fusion-spliced an SMF with a hollow capillary tube and coated the end of the capillary tube with a layer of polyimide film. The film and the end-face of the SMF formed the FP cavity, and an FBG was cascaded to eliminate the influence of temperature changes. When the external humidity changed, the polyimide underwent deformation [111], resulting in a change in the length of the FP cavity and a shift in the interference spectrum. Experimental results indicated that the sensitivity of this sensor in the range of 20% to 90% relative humidity (RH) was 22.07 pm/%RH. Reference [112] used polyvinylidene fluoride (PVDF) as the humidity-sensing film. The structure and sensing principle was similar to the previous approach. The sensor exhibited a sensitivity of 32.54 pm/%RH at a constant temperature. Furthermore, reference [113] proposed a humidity sensor based on polyvinyl alcohol (PVA) film. The author fusion-spliced a section of hollow-core fiber between two segments of SMFs and coated the other end of the second segment of SMF with PVA film. The end-faces of the two aligned SMFs formed $FP_1$, and the two ends of the second segment of SMF formed $FP_2$. When the external humidity changed, the PVA film underwent deformation [114], causing a change in the length of the $FP_2$ cavity and resulting in a spectral shift. The cascaded FP cavities generated a vernier effect, enhancing the sensitivity of the sensor [115]. The author also mentioned that the envelope spectrum of the sensing cavity and the two FP cavities can be separated using a bandpass filter, effectively addressing the issue of cross-sensitivity between temperature and humidity. Experimental results indicated that when the humidity increased from 35% RH to 85% RH, the sensitivity to humidity reached 1.454 nm/%RH.

Besides polymers for humidity sensing, there are other materials that can be used in fiber optic FP cavity humidity sensors. Reference [116] proposed a fiber optic FP humidity sensor based on porous anodic alumina (PAA) film. The author used UV adhesive to attach the porous PAA film to the end of an SMF and the capillary structure of the film caused changes in the effective refractive index due to the condensation of water molecules [117]. Experimental results showed that the sensitivity in the range of 20% to 90% RH could reach 0.31 nm/%RH. Reference [118] presented an FP cavity utilizing the hydrophilic properties of graphene oxide. The author fusion-spliced a capillary tube to the end of an SMF and added a graphene oxide diaphragm at the end of the capillary tube. Due to the absorption and desorption behavior of water molecules by graphene oxide, it underwent expansion or contraction [119], resulting in a change in the length of the FP cavity and a shift in the interference spectrum. Experimental results demonstrated that this humidity sensor exhibited an average wavelength variation of 0.2 nm/%RH in the range of 10% to 90% RH. The performance of fiber optic FP humidity sensors based on different film is summarized in the Table 4 in this section.

**Table 4.** Humidity sensor performance based on different materials.

| Material | Sensitivity (nm/%RH) | Range (%RH) | Reference |
|---|---|---|---|
| PMMA | 0.1747 | 25~80 | [101] |
| | 0.127 | 35~85 | [102] |
| | 0.4172 | 10~70 | [102] |
| PAM | 0.1 | 38~78 | [104] |
| | 5.868 | 88~98 | [104] |
| Agarose gel | 0.0225 | 43~63 | [106] |
| PNIPAM | 1.634 | 45~75 | [108] |
| POLYIMIDE | 0.02207 | 20~90 | [110] |
| PVDF | 0.03254 | 20~80 | [112] |
| PVA | 0.001454 | 35~85 | [113] |
| PAA | 0.31 | 20~90 | [116] |
| Graphene oxide | 0.2 | 10~90 | [118] |

*3.5. Gas Detection*

Gas detection plays a crucial role in various aspects of everyday life, such as ensuring the safety of spaces, environmental protection, industrial production, health preservation, and disaster response. In recent years, fiber optic sensing has achieved significant advancements in gas detection. Examples include gas sensors based on fiber Bragg gratings [120,121], Sagnac interferometry [122,123], Mach–Zehnder interferometry [124,125], and Michelson interferometry [126,127]. This section will introduce fiber optic FP Interferometer (FPI) gas detection sensors based on different thin films.

Carbon dioxide ($CO_2$) is one of the major greenhouse gases closely associated with global climate change. Monitoring $CO_2$ concentration provides valuable information about the accumulation of greenhouse gases in the atmosphere. This is crucial for assessing the impact of climate change, formulating mitigation and adaptation measures, and promoting sustainable development. In reference [128], a $CO_2$ sensor based on a polyethyleneimine/poly (vinyl alcohol) (PEI/PVA) coating was proposed. The authors coated the end-face of an SMF with PEI/PVA, forming an FP cavity between the end-faces of the SMF and the PEI/PVA coating. When the PEI/PVA film is exposed to varying $CO_2$ environments, it can alter the optical path difference of the FPI [129]. Consequently, the $CO_2$ gas concentration can be measured through wavelength shifts in the interference fringe pattern. The proposed FPI sensor exhibited high sensitivity to changes in $CO_2$ concentration, with a sensitivity of 0.281 nm/% in the range of 7.6% to 86.9%. In reference [130], a $CO_2$ sensor utilizing polyhexamethylene biguanide (PHMB) as the thin film was presented. The authors coated the end of an SMF with PHMB film as the reflective coating for the FP cavity. Absorption and release of $CO_2$ gas molecules caused variations in the refractive index of PHMB, leading to spectral shifts in the interference pattern [131]. Experimental results demonstrated a sensitivity of 1.22 pm/ppm in the $CO_2$ concentration range of 0–700 ppm. However, both above-mentioned sensors exhibited a response to temperature, necessitating further improvements to address cross-sensitivity. Techniques such as incorporating FBG into the SMF can be employed to solve this issue.

As is well known, carbon monoxide (CO) is an explosive and toxic gas that, when inhaled, can cause oxygen deprivation in the blood, leading to harm to the heart and nervous system and even coma or death. However, its colorless and odorless nature makes it difficult to detect, highlighting the necessity of highly sensitive carbon monoxide sensors. Reference [132] proposed and fabricated an optical fiber FP interferometric carbon monoxide gas sensor based on a polyaniline/$Co_3O_4$ (PANI/$Co_3O_4$) film. The sensor consisted of a segment of SMF fusion-spliced with an endlessly photonic crystal fiber (EPCF), with the PANI/$Co_3O_4$ thin film coated on the end of the EPCF. This film adsorbed CO molecules, thereby altering the optical path of light transmission within the film.

Experimental results showed that within the range of 0–70 ppm, the sensor achieved a sensitivity of 21.61 pm/ppm for detecting CO. The authors further immersed the sensor in air containing various gases such as nitrogen, carbon dioxide, and oxygen, which are commonly present in the atmosphere. They observed that the interference spectrum did not show any significant changes, indicating the sensor's high selectivity towards carbon monoxide. However, it was noted that the sensor exhibited relatively long response and recovery times, which could potentially be attributed to the characteristics of the materials used in the sensor design. Further investigation is needed to improve the response and recovery times of the sensor.

Ammonia ($NH_3$) detection has broad practical significance, including industrial safety, agricultural applications, indoor air quality, and biological research. Reference [133] compared fiber optic FPI gas sensors based on porous graphene film (G-FPI) and $Fe_3O_4$-graphene nanocomposite film (FG-FPI) for the detection of ammonia gas at room temperature. The mechanism of these two sensors was based on the change in the refractive index of the thin film under different $NH_3$ gas concentrations. The sensor structure was simple, with a sensing film coated on the end of an SMF. However, both sensors initially exhibited good linearity at low $NH_3$ concentrations, but as the concentration increased, a nonlinear relationship appeared due to saturation effects. The sensors were tested in the range of 1.5 ppm to 150 ppm, and the experimental results showed that the G-FPI sensor achieved a sensitivity of ~25 pm/ppm at 150 ppm, while the FG-FPI sensor achieved a sensitivity of ~36 pm/ppm, the lowest detection limit of both sensor probes could be in the range of around 10 ppb and 7 ppb, respectively. Throughout the measurement process, the FG-FPI sensor outperformed the G-FPI sensor. Reference [134] investigated the influence of the thickness of ITO and $SnO_2$ on the sensitivity of microstructured optical fiber FP sensors. The authors connected a section of four-bridge double-Y-shape core microstructured optical fiber (MOF) to the end of an SMF to form an FP cavity and deposited metal oxides inside the holes of the MOF. The sensing mechanism was based on the change in the refractive index of the metal oxide after gas molecule adsorption [135]. However, both oxide-based sensors exhibited relatively long response and recovery times. The performance of fiber optic FP gas sensors based on different thin films is summarized in Table 5 in the original paper.

**Table 5.** Gas detection sensor performance based on different materials.

| Material | Sensitivity (pm/ppm) | Range (ppm) | Reference |
|---|---|---|---|
| PEI/PVA | 0.281 | 76,000~869,000($CO_2$) | [128] |
| PHMB | 1.22 | 0–700 ($CO_2$) | [130] |
| PCG | 21.61 | 0~70 (CO) | [132] |
| G-FPI | 25 | 0~150 ($NH_3$) | [133] |
| FG-FPI | 36 | 0~150 ($NH_3$) | [133] |

### 3.6. Temperature Sensor

Temperature measurement has important applications in various fields, such as industrial production, construction, healthcare, and nuclear energy. However, existing electronic temperature sensors are difficult to use in the presence of strong electromagnetic fields. In contrast, optical fibers can overcome this interference, as demonstrated by temperature sensors based on FBG [136,137], WGM [138,139], MZI [140,141], and Michelson interferometry [142,143]. This section will focus on the recent achievements of fiber FP devices based on different thin films for temperature sensing.

A fiber FP sensor based on silicon thin film was proposed in [144] for simultaneous temperature and pressure measurements. The author inserted an SMF into a glass ferrule, followed by the adhesion of heat-resistant glass and silicon wafer. The silicon wafer had a silicon film at its end to sense external temperature and pressure changes, while the heat-resistant glass only responded to temperature. By using bandpass filtering, the signals provided by the two sensing elements could be extracted separately, addressing the issue of cross-sensitivity between temperature and pressure. Experimental results showed a

sensitivity of 142.02 nm/°C within the range of −20 to 70 °C for this sensor. Another temperature sensor based on a silicon thin film was presented in [145], as shown in Figure 7. The author inserted a multimode fiber into a glass ferrule and fused an empty cavity at the end, which was then bonded with a layer of silicon film. This silicon film divided the cavity into two chambers, with the chamber closer to the multimode fiber serving as the FP cavity and the other side as the sensing chamber. As the external temperature increased, the thermal expansion of the air inside the sensing chamber caused a pressure difference between the two chambers, leading to the deformation of the silicon film and resulting in a change in the cavity length of the FP cavity. Experimental results demonstrated a temperature sensitivity of 6.07 nm/°C within the range of −50 to 100 °C, and the response time of the sensor within the range of 28 to 100 °C was approximately 1.3 s.

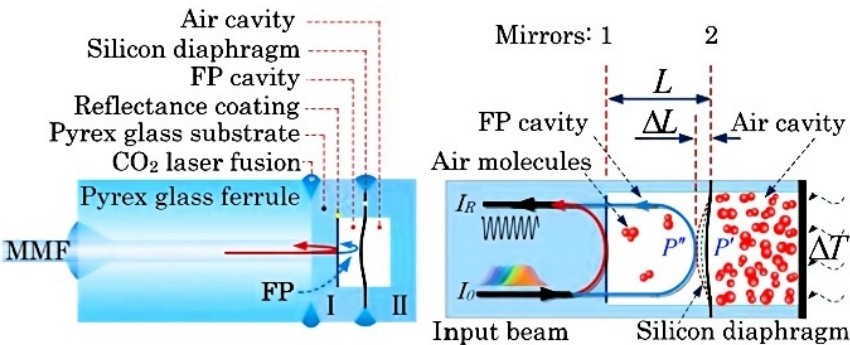

**Figure 7.** Differential-pressure-based fiber optic temperature sensor [145].

In addition to silicon thin film, alternative materials are also employed for temperature sensing in fiber FP sensors. An optic fiber FP temperature sensor based on PDMS thin film was proposed in reference [146]. The sensor consisted of two sensing devices, with each FP cavity formed by splicing a section of silica tube to the end of an SMF. One cavity was filled with a PDMS thin film, denoted as $FP_1$, which was sensitive to both temperature and pressure, while the other cavity was filled with a layer of UV adhesive thin film, denoted as $FP_2$, which was only sensitive to pressure. Like the previous approach, the use of dual FP cavities generated a vernier effect to enhance the sensing sensitivity. By employing bandpass filtering to extract the signals from each FP cavity separately, the issue of cross-sensitivity between temperature and pressure could be resolved. The temperature sensitivity of this sensor within the range of 44 to 49 °C was measured to be 10.29 nm/°C. An FPI based on a Polyimide tube for measuring seawater temperature and pressure was proposed in reference [147], as shown in Figure 8a. In the FP cavity, variations in external temperature and pressure caused the deformation of the PI tube, leading to a change in the FP cavity length. To address the issue of temperature and pressure cross-sensitivity, a cascaded FBG was employed. The higher thermal expansion coefficient of the polymer compared to silicon-based devices contributed to improved sensor sensitivity. Experimental results indicated a temperature sensitivity of 18.910 nm/°C within the range of 24 to 43 °C. In [148], PDMS was deposited on the end of a standard SMF, and it included a layer of carbon nanoparticles (CNPs). The optical-thermal effect of carbon nanoparticles [149] was utilized to create a microbubble within the coating using a laser, enhancing the sensitivity of the sensor. The structure of this sensor is depicted in Figure 8b. The temperature sensitivity of this sensor reached up to 790 pm/°C within the temperature range of 27 to 40 °C. In reference [150], the authors employed a rapid immersion method of multimode fiber into molten tellurite glass, which alleviated the challenge of material fusion. Upon solidification of the tellurite glass, a microcavity is formed at the end of the multimode fiber, serving as an FP cavity. The external temperature variations induce changes in the refractive index and shape of the tellurite glass, resulting in the shift of the interference spectrum. The temperature sensitivity of the sensor reaches 62 pm/°C within a wide measurement range of 20 °C to 170 °C. Furthermore, this sensor demonstrates fast response and excellent stability.

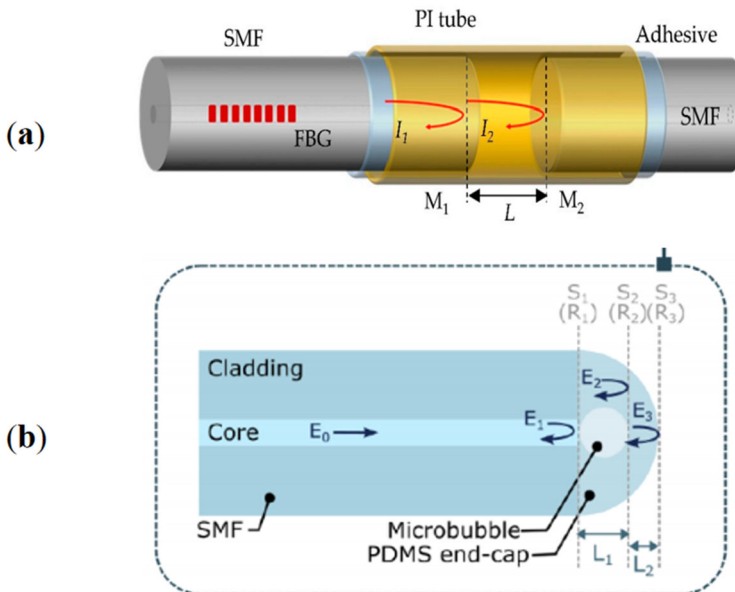

**Figure 8.** (**a**) Temperature sensors based on PI (polyimide) tube [147] (**b**) Temperature sensors based on PDMS film [148].

Solid materials often exhibit uneven thermal stress during the sensing process, which can increase the response time of the sensor [151]. On the other hand, liquid materials rarely experience thermal stress imbalance, allowing for faster sensor response. In reference [152], the authors constructed an FP sensing cavity by connecting an SMF, a thick-core capillary, a thin-core capillary, and another SMF in sequence. The capillary was filled with ethanol solution. When the temperature rises, the ethanol solution expands, causing a change in its refractive index [153], which leads to a shift in the interference spectrum. Since the fusion splicing process involves discharging, some of the ethanol solution vaporizes at high temperatures, resulting in an air column. The article points out that this sealed space increases the boiling point of ethanol, so the air column should not be too large. However, it should not be too small either, as it would restrict the expansion of ethanol, affecting the sensitivity of the sensor. Experimental data show that when the volume ratio of ethanol to air is 2.89, the temperature sensitivity of the sensor reaches −497.6 pm/°C within the range of 20 to 180 °C, and the maximum detectable temperature can reach 220 °C. Furthermore, the proposed fiber temperature sensor exhibits fast response (less than 1 s for temperature variations of about 90 °C) and good repeatability.

The previously mentioned fiber temperature sensors have a maximum temperature limit of only 220 °C, which is insufficient for high-temperature measurements. In [154], a fiber FP temperature sensor using sapphire material was proposed. The authors fixed a sapphire chip on the end-face of a sapphire fiber as the FP cavity. When the temperature changed, the thickness and refractive index of the sapphire chip changed, resulting in a shift in the interference spectrum. The sensor was tested within the range of 25 to 1550 °C, and the sensitivities at temperatures of 500, 1000, and 1550 °C were 20.63, 26.25, and 32.45 pm/°C, respectively. Another sapphire fiber FP structure for high-temperature measurement was proposed in reference [155]. The FP cavity was formed by the end-face of the sapphire fiber and the end-face of a sapphire rod. The sensor was experimentally tested within the range of 0 to 1500 °C, but the data showed that the sensor's response to temperature was nonlinear, and the sensitivity increased with temperature.

In the field of low-temperature measurements, there are three commonly used instruments: helium vapor pressure thermometers, gas thermometers, and platinum resistance thermometers. At low temperatures, the thermal expansion coefficient of silica (SiO2) is extremely low, rendering bare fiber Bragg gratings (FBGs) unsuitable for temperature sensing below 40 K [156]. Therefore, additional materials are needed to enable fiber-based

temperature sensing at low temperatures, such as the application of metal coatings on FBGs. Among them, lead-coated FBGs are currently the most sensitive, exhibiting a sensitivity of 8.7 pm/K at 5 K [157]. In reference [158], a fiber FP sensor designed for low-temperature measurements was proposed. The sensor featured a simple structure, where two SMFs were enclosed within ceramic rings, forming an FP cavity with the fiber end-faces. The assembly was then inserted into a copper sleeve. The working principle was straightforward: as the external temperature changed, both the ceramic rings and the copper sleeve underwent deformations proportional to the temperature variations, causing a shift in the interference spectrum. The copper sleeve was crucial for the sensor's low-temperature sensing range of 5–75 K since copper maintained a considerable thermal expansion coefficient within this range, whereas the thermal expansion coefficients of the ceramic rings (alumina) and silica started to increase gradually from near-zero to around 75 K. The study also conducted a further analysis of the sensor and suggested that zinc, with a thermal expansion coefficient better than copper below 50 K, could potentially be employed as a substitute to enhance sensing performance. In summary, this paper presents a rare investigation of fiber FP sensors in low-temperature conditions. Roctest Company offers two FP-based fiber optic temperature sensors, namely FOT-F and FOT-N. These sensors are designed based on highly stable glass materials that exhibit thermal expansion. They are suitable for highly precise, stable, and repeatable measurements. FOT-F can be used in vacuum environments, high-pressure applications, or high-voltage environments, while FOT-N is primarily used for embedding in concrete structures or air. The performance of the aforementioned fiber FP temperature sensors for temperature measurement is summarized in Table 6.

**Table 6.** Temperature sensor performance based on different materials.

| Material | Sensitivity | Test Range | Reference |
|---|---|---|---|
| Silicon | 142.02 nm/°C | −20~70 °C | [144] |
| | 6.07 nm/°C | −50~100 °C | [145] |
| PDMS | 10.29 nm/°C | 44~49 °C | [146] |
| | 62 pm/°C | 20~170 °C | [148] |
| PI | 18.910 nm/°C | 24~43 °C | [147] |
| Ethanol | −497.6 pm/°C | 20~180 °C | [152] |
| Sapphire | 20.63 pm/°C (at 500 °C) | 25~1550 °C | [154] |
| | 26.25 pm/°C (at 1000 °C) | 25~1550 °C | [154] |
| | 32.45 pm/°C (at 1550 °C) | 25~1550 °C | [154] |
| Cu/Al$_2$O$_3$ | 2.10 nm/K | 5.367~15.069 K | [158] |
| | 1.95 nm/K | 15~50 K | [158] |
| | 7.73 nm/K | 96.5~142.69 K | [158] |
| | 5.33 nm/K | 150.19~200.36 K | [158] |
| | 4.35 nm/K | 250.18~290.98 K | [158] |

*3.7. Biological or Medical Sensor*

In recent years, fiber optic sensing has attracted increasing attention in the fields of biology and medicine, such as the use of LPG-based biosensors [159,160], Mach–Zehnder Interferometer (MZI)-based medical sensors [161,162], and SPR-based biosensors [163,164]. This section introduces some research on fiber optic FP sensors in the field of biology and medicine. Reference [165] proposed a fiber optic FP sensor for detecting Microcystin-LR (MCT). The authors coated the end-face of an SMF with a Molecularly Imprinted Polymer to form the FP cavity, which exhibits affinity for the selected "template molecule". When the sensor meets the MCT solution, the refractive index changes, leading to a shift in the interference spectrum. Experimental results showed that the sensor has a sensitivity of approximately 12.4 nm L/μg in the concentration range of 0.3–1.4 μg/L for MCT. Reference [166] presented a fiber optic FP interferometric immunosensor based on chitosan/polystyrene sulfonate film, as shown in Figure 9. Chitosan is a biopolymer that

can preserve the biological properties of proteins, making it suitable for immobilizing immunoglobulin G (IgG). When anti-immunoglobulin G (anti-IgG) is adsorbed onto the chitosan substrate, the adsorbed protein can be considered to be forming a membrane, which increases the thickness of the sensing film and adjusts its refractive index, resulting in the shift of the output interference spectrum. The focus of the experiment was on the power variation in the spectrum. The authors determined protein binding events by demodulating the changes in effective optical length. The sensitivity of the sensor is 0.033 m/(pg/mm$^2$). Reference [167] proposed a fiber optic FP sensor based on antibody–antigen-specific binding. The authors prepared the FP interferometer by splicing the ends of a short-section hollow-core photonic crystal fiber to an SMF and cutting the SMF pigtail to an appropriate length. Then, goat anti-rabbit immunoglobulin G (IgG) was covalently immobilized and fixed onto the salinization-modified end of an SMF to detect the specific rabbit IgG. The specific binding of anti-rabbit IgG and rabbit IgG changed the thickness of the sensing layer, causing a change in light absorption by the reflection surface fixed by the biomolecular layer and a change in the effective cavity length of the FP sensor, resulting in a change in the optical intensity and a shift in the interference spectrum. Experimental results showed that during the antigen binding process, the wavelength increased by 190 pm, and the fringe contrast decreased by 2.15 dB, indicating the successful binding of the antibody and antigen, thereby confirming the feasibility of the proposed immunosensor.

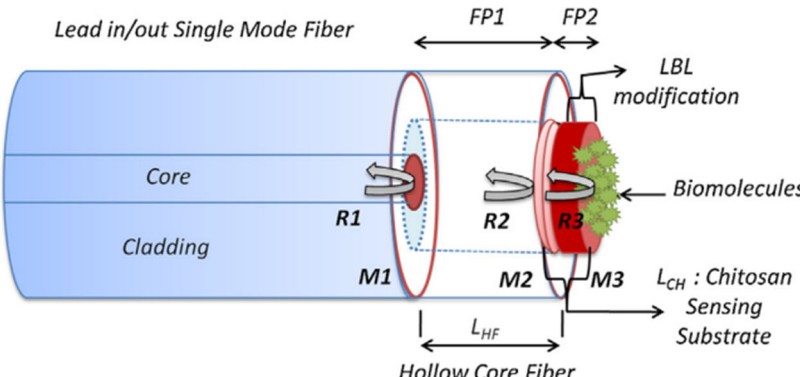

**Figure 9.** F-P interferometric immunosensor based on a chitosan/polystyrene sulfonate film [166].

Reference [168] presented a fiber optic FP sensor based on polypropylene film for respiratory detection in humans. This thin film had a low Young's modulus, which enhanced pressure-sensing sensitivity. The sensor structure was also simple, with a polypropylene film prepared at the end of a nut and an SMF inserted at the other end to form the FP cavity. Experimental data showed a sensitivity of −0.581 nm/Pa for the respiratory sensor. However, this sensor was influenced by humidity, particularly in high humidity conditions, which limited its operating conditions. Reference [169] proposed a fiber optic FP sensor for monitoring human respiration using humidity sensing. The sensing mechanism utilized the adsorption of water molecules by chitosan polymers, which caused changes in refractive index and volume [170]. The authors fusion-spliced a section of MOF (Microstructured Optical Fiber) to the end of an SMF to form the FP cavity and filled the cavity with chitosan polymer. Experimental results showed that within the range of 70% to 95% relative humidity (RH), the sensor has a sensitivity of 68.55 pm/%RH. The impact of temperature within the range of 20 to 70 °C is minimal. This sensor exhibits fast response times, with a rise time of 80 ms and a recovery time of 70 ms. Reference [171] proposed a fiber optic sensor for DNA detection. Two sections of C-type optical fibers were inserted into an SMF to form two FP cavities. One FP cavity was used for DNA detection, where the binding of pDNA and cDNA in the solution adds a biological layer on the cross-section of the SMF, resulting in a refractive index change and a shift in the interference spectrum. The other FP cavity was filled with PDMS to monitor environmental temperature changes and prevent interference during the detection process.

Reference [172] presented a fiber optic FP sensor for measuring blood temperature. The authors fusion-spliced a short section of multimode fiber to the end of an SMF and etched a groove to place borosilicate glass, forming the FP cavity. Due to the high coefficient of thermal expansion of borosilicate glass, the FP cavity length changed when the external temperature varied, resulting in a shift in the interference spectrum. Experimental results showed that the sensor has a sensitivity of 0.0103 nm/°C within the range of 38 to 40 °C. This sensor is flexible for temperature measurement in various locations of the body, and it exhibits a very fast response time of less than 1 s. Reference [173] proposed a fiber optic FP sensor for monitoring blood pressure change. The authors fusion-spliced a section of multimode fiber to the end of an SMF and etched a groove at the end of the multimode fiber. They then bonded a thin film of silicon dioxide to the end for pressure sensing. When the external pressure changed, the deformation of the thin film led to a change in cavity length, and the blood pressure could be determined based on the spectral shift. Experimental tests were conducted in pig arteries, and the sensor demonstrated a sensitivity of 0.035 mV/mmHg. Reference [174] introduced a fiber optic FP sensor for monitoring heart rate. The authors inserted two sections of SMFs into a capillary to form the FP cavity and bonded the SMFs to the capillary using ethyl cyanoacrylate (EtCNA). The device was then placed within a circular frame. The material EtCNA has a low Young's modulus, making it suitable for detecting low-frequency vibrations. When external vibrations occurred, the circular frame underwent deformation, resulting in a change in the cavity length of the interferometric device. Experimental results demonstrated that the sensor has a strain sensitivity of 2.57 pm/μN and exhibited good response to low-frequency vibrations at 1~3 Hz. Currently, there are several fiber optic FP sensors available on the market for medical applications. For instance, the FOP-M260(FISO Technologies, Inc., Quebec, QC, Canada) fiber optic pressure sensor is specifically designed for the medical field. It is a small-sized, high-precision sensor that can be used for monitoring left ventricular pressure, arterial blood pressure, intracranial pressure, and other parameters. The sensor ensures high stability and does not pose any harm to the human body.

## 4. Conclusions

This paper presents a comprehensive overview and analysis of fiber FP sensors based on different thin films. In various sensing applications, it is crucial to select thin film materials that exhibit sensitivity to the specific physical quantities being measured, either as the core components of the sensor or to enhance its sensitivity. Challenges encountered in practical implementation include achieving secure integration of thin film with optical fibers and precise control over film size and thickness to meet the requirements of high-performance sensors, thereby enabling the fabrication of stable and reliable fiber FP sensors. Additionally, addressing the issue of mutual interference among multiple parameters and cross-sensitivity, where a particular material is influenced by more than one physical quantity, represents an important research direction for fiber FP cavity sensors. In summary, thin film technology and FP cavities, as essential components of fiber optic sensing, offer novel avenues to enhance the sensitivity, selectivity, and reliability of sensors. With continued technological advancements and expanded applications, this fiber optic sensing technology holds great promise for playing a more significant role in various fields and providing innovative solutions to scientific and engineering problems.

**Author Contributions:** Conceptualization, S.L.; methodology, C.M. and D.P.; formal analysis, X.B.; writing—original draft, C.M. and D.P.; writing—review and editing, S.L.; supervision, S.L. and L.L. All authors have read and agreed to the published version of the manuscript.

**Funding:** National Key Research and Development Program (Grant No. 2022YFC2204402), Guangdong Science and Technology Project (Grant No. 20220505020011), Shenzhen Science and Technology Program (Grant No. 2021Szvup172), and Shenzhen Science and Technology Program (Grant No. JCYJ20220818102003006).

**Institutional Review Board Statement:** Not applicable.

**Informed Consent Statement:** Not applicable.

**Data Availability Statement:** Data-sharing is not applicable to this article.

**Conflicts of Interest:** The authors declare no conflict of interest.

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
