# Peer review of "A Review of Optical Fiber Sensing Technology Based on Thin Film and Fabry–Perot Cavity"

_coatings, doi:10.3390/coatings13071277_

Round 1
Reviewer 1 Report
Authors should revise the Manuscript based on the following comments.
1. Importance of Fabry-Perot cavity in the sensor performance must be explained.
2. various fiber sensors are available in the literature. Since authors focused on FP cavity, they need to role of Fabry-Perot cavity in the fiber sensor technology.
3. Transmission spectrum with fiber length is very important for fiber sensors. Discuss it.
4. Figure of merit and Limit of detection are important sensing parameters for fiber sensors. Authors need to discuss about them.
5. Justify, how your method is novel as compared to the literature.
6. It is observed that there are many grammatical mistake in many places and it is suggested to take proof reading with English master before submitting next version.
Author Response
Thank you very much for your comments on our earlier manuscript. Those comments are all valuable and very helpful for revising and improving our paper, as well as the important guiding significance to our research.
We have rewritten the manuscript according to your comments, and now we resubmit it to this journal again. We hope the manuscript has been improved satisfactorily and the quality of our manuscript would meet the publication standard of Coatings.
Our responses to your comments are as follows. The red font was used in the revised manuscript for the changes made in comments 1 to 5, and changes made to comment 6 were highlighted. In addition, changes according to other reviewers were marked in blue and green respectively.
Thank you very much for your time and effort that goes into the modification of this paper.

Reviewer 2 Report
Optical fiber sensors based on thin film membranes and Fabry-Perot (FP) cavity represent a wide class of optical sensing microdevices, which have a huge potential due to their ability to determine multiple physical quantities with high precision.
The manuscript presents an excellent comprehensive review on design principles, detection mechanisms, performance indicators, recent practical developments, and various applications of FP fiber sensors. In my opinion the paper is generally well written and well structured, the list of references is adequate to the current state of this field of research and technology, but some important points need improvement and clarification.
1. All statements made in the Introduction should be supported by appropriate references.
2. Any previous reviews on this topic (FP fiber sensors) were not mentioned in the manuscript. What new contributions in terms of overviewing recent publications or comprehensive and systematical analysis were made by the authors compared to other similar reviews?
3. FP fiber sensors for measuring various physical quantities (pressure, humidity, temperature etc.) should be put in a broader context, their performance characteristics should be compared to other types of fiberoptic sensors designed for the same task.
4. For chemical gas sensors (Section 3.5, Table 5) alongside with sensitivity it might be useful to add to Table 5 the information about limits of detection (LOD) for the described sensors if such data are available in the original papers.
Author Response
Thank you very much for your comments on our earlier manuscript. Those comments are all valuable and very helpful for revising and improving our paper, as well as the important guiding significance to our research.
We have rewritten the manuscript according to your comments, and now we resubmit it to this journal again. We hope the manuscript has been improved satisfactorily and the quality of our manuscript would meet the publication standard of Coatings.
Our responses to your comments are as follows. Revised portions are marked in blue in the revised manuscript, while changes according to other reviewers were marked in red and green respectively. In addition, we improved our expression to prevent some grammatical mistakes, and all changes were highlighted.
Thank you very much for your time and effort that goes into the modification of this paper.

Reviewer 3 Report
The paper “A Review of Optical Fiber Sensing Technology Based on Thin Film and Fabry-Perot Cavity” by C.Ma, et al contains a large amount of material devoted to specific fiber sensors, namely fiber sensors using different types of thin films forming Fabry-Perot cavity for readout of temperature, pressure and other physical parameters.
Since the material is presented in the form of a review, it makes sense to evaluate its completeness. In principle, I think that the text gives a fairly complete overview of this specific type of sensor. Nevertheless, I would like to make a few remarks of a recommendatory nature.
- It would make sense to present summary data on the dynamics of the number of publications on this material over the past few years.
- Since we are talking about sensors, it would be recommended to present examples commercial implementations of such sensors.
- There is another type of fiber sensor using Fabry-Perot resonances, namely photonic crystal fibers. Of course, the geometry of such sensors is different than . But maybe it would make sense to discuss the advantages/disadvantages.
Author Response
Thank you very much for your comments on our earlier manuscript. Those comments are all valuable and very helpful for revising and improving our paper, as well as the important guiding significance to our research.
We have rewritten the manuscript according to your comments, and now we resubmit it to this journal again. We hope the manuscript has been improved satisfactorily and the quality of our manuscript would meet the publication standard of Coatings.
Our responses to your comments are as follows. Revised portions are marked in green in the revised manuscript, while changes according to other reviewers were marked in red and blue respectively. In addition, we improved our expression to prevent some grammatical mistakes, and all changes were highlighted.
Thank you very much for your time and effort that goes into the modification of this paper.

Round 2
Reviewer 1 Report
Revised paper can be accepted
Reviewer 2 Report
The manuscript has been improved.